# LMM-PCQA: Assisting Point Cloud Quality Assessment with LMM

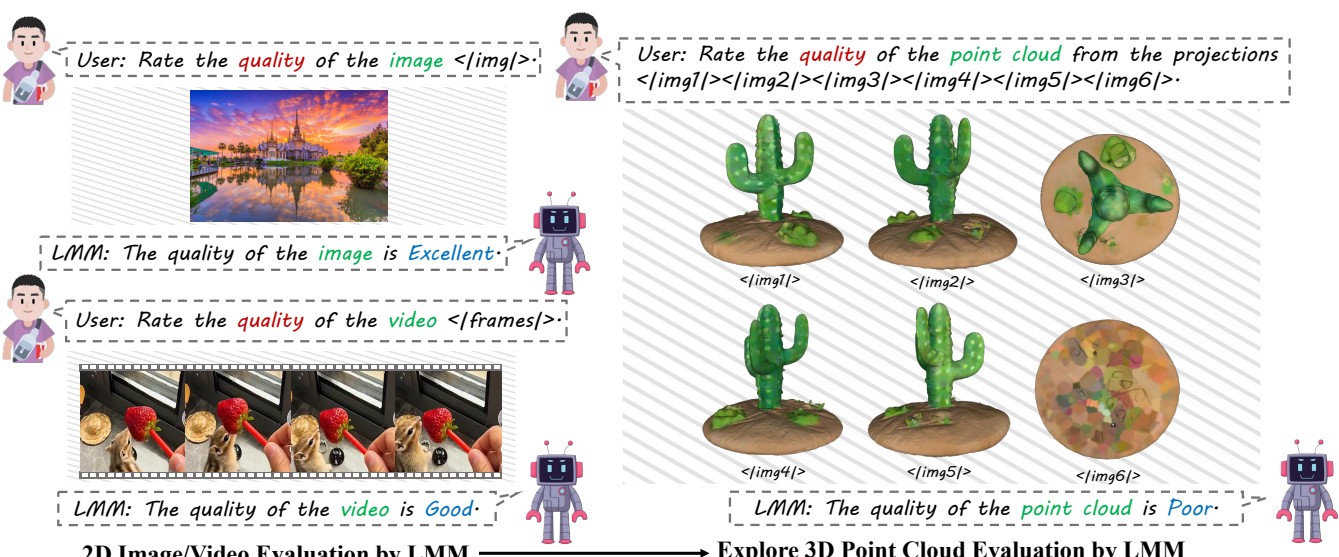

**Figure 1: Inspired by the impressive quality evaluation ability of LMM on 2D media, we are the first to explore the quality representation potential of LMM on 3D point clouds.**

## ABSTRACT

Although large multi-modality models (LMMs) have seen extensive exploration and application in various quality assessment studies, their integration into Point Cloud Quality Assessment (PCQA) remains unexplored. Given LMMs' exceptional performance and robustness in low-level vision and quality assessment tasks, this study aims to investigate the feasibility of imparting PCQA knowledge to LMMs through text supervision. To achieve this, we transform quality labels into textual descriptions during the fine-tuning phase, enabling LMMs to derive quality rating logits from 2D projections of point clouds. To compensate for the loss of perception in the 3D domain, structural features are extracted as well. These quality logits and structural features are then combined and regressed into quality scores. Our experimental results affirm the effectiveness of our approach, showcasing a novel integration of LMMs into PCQA that enhances model understanding and assessment accuracy. We hope our contributions can inspire subsequent investigations into the fusion of LMMs with PCQA, fostering advancements in 3D visual quality analysis and beyond.

## CCS CONCEPTS

• **Human-centered computing** → *Visualization design and evaluation methods*; • **Computing methodologies** → **Artificial intelligence**.

## KEYWORDS

Large multi-modality model, Point cloud quality assessment

## 1 INTRODUCTION

Point clouds are increasingly used across diverse real-world scenarios, including virtual/augmented reality [15, 21, 31], autonomous vehicles [9], and video post-production [26]. This surge is attributed to their adeptness in three-dimensional representation. Consequently, significant research efforts have been channeled towards enhancing high-level areas like point cloud classification [5, 13, 17, 37, 39, 44, 51], detection [9], and segmentation [7, 24]. Meanwhile, as a key component for ensuring the point cloud quality, point cloud quality assessment (PCQA) has seen comparable advancements as well during the last decade. PCQA's objective is to appraise point clouds' visual quality, a pivotal factor for refining simplification and compression strategies in practical applications [10], and to elevate the Quality of Experience (QoE) for end-users. The complexity of PCQA is further compounded when assessing point clouds depicting realistic objects or individuals, due to their intricate geometric structures and the dense aggregation of points, often augmented with color attributes, posing additional challenges. Generally, PCQA methods are divided into three categories, depending on their reliance on reference point clouds: Full-Reference

PCQA (FR-PCQA), Reduced-Reference PCQA (RR-PCQA), and No-Reference PCQA (NR-PCQA) respective. The availability of pristine reference point clouds is often limited in real-world applications, emphasizing the demand for NR-PCQA approaches, which is why our research primarily focuses on NR-PCQA.

There are already some cutting-edge studies that have begun applying Large Multi-Modality Models (LMMs) to low-level vision and quality assessment fields [16, 40–42, 52], achieving notable success. LMMs demonstrate highly competitive performance and robustness in these tasks. However, the main focus of these studies has remained on two-dimensional (2D) media such as images and videos, **while no research has explored the possibility of applying LMMs to three-dimensional (3D) media like point clouds.** It is known that both 2D and 3D media exhibit similar distortions, i.e., blur and noise. Given the robust quality perception of LMMs in 2D, **we can hold the hypothesis that LMM also has significant quality perception abilities in 3D point clouds.** Hence, investigating the application of LMMs for point cloud quality assessment is not only valuable but also meaningful, reflecting their established visual perception strengths. Therefore, in this study, we carry out a novel method named **LMM-PCQA** to provide an interesting solution for handling the PCQA problem with the assistance of LMM.

First, **we treat the point clouds as sequences of projections** to enable LMM to perceive the point cloud visual quality. Afterward, we try to **teach LMM about the quality alignment** between the predefined 5-level **qualitative adjectives** (i.e., *excellent, good, fair, poor, bad*) and point cloud projections. Specifically, we employ the existing PCQA databases to provide the necessary knowledge, where the quality labels are transformed into corresponding **qualitative adjectives**. Then we specially design a prompt structure to produce the question-answer pairs, which are composed of the question *'Rate the quality of the point cloud from the projections [img1],[img2],[img3],[img4],[img5],[img6]'* and the answer *'The quality of the point cloud is excellent/good/fair/poor/bad'*. These question-answer pairs can be utilized to teach LMM PCQA knowledge during the fine-tuning stage. After the instruction tuning, we can expect the trained LMM to give the predicted *[SCORE_TOKEN]* with the same prompt structure (the **qualitative adjectives** position is left blank for LMM response), which is shown in Fig. 2. The predicted *[SCORE_TOKEN]* can be recognized as a probability map to the **qualitative adjectives**, and we convert it into 5-level probabilities as the LMM evaluation results.

Secondly, to address the potential insensitivity to geometric distortions (e.g., compression, downsampling) when only analyzing projections, **we propose the extraction of multi-scale structural features.** This approach enhances the LMM-PCQA's holistic comprehension of point cloud visual quality. The point clouds are converted into quality-aware structural domains, a technique validated for effective quality feature extraction in prior research [3, 58, 65]. We modify the scale parameters in the k-nearest neighbors (k-NN) algorithm to offer a multi-scale perspective, aligning with the human vision system's perception mechanism. Subsequently, key statistical parameters are used to quantify structural distortions within these domains. Finally, we combine the LMM's evaluative results and the structural features, utilizing support vector regression (SVR) to derive the quality values. The experimental

outcomes affirm that our LMM-PCQA model is on par with, or surpasses, current leading PCQA methods.

By carrying out LMM-PCQA, the contributions of this paper can be summarized as follows:

- **We are the first to employ LMM for PCQA tasks.** We design a novel prompt structure to enable the LMM to perceive the point cloud visual quality. The existing PCQA databases are converted into question-answer pairs, which are then used to **inject PCQA knowledge into LMM**.
- **We propose the extraction of multi-scale structural features.** By processing point clouds into multi-scale domains, we quantify the geometry distortions via key statistic parameters estimation, which helps LMM gain a more comprehensive understanding of point cloud visual quality.
- **LMM-PCQA demonstrates exceptional performance across various PCQA databases.** The ablation study and cross-database evaluations further validate the logical design of LMM-PCQA and its robust generalization capabilities.

## 2 RELATED WORKS

### 2.1 LMM for Quality Assessment

Recent studies have begun to explore the utilization of LMMs for visual quality assessment, marking a significant shift in the field. For instance, X-IQE [6] employs LMMs to assess the quality of text-to-image generation methods, leveraging a chain of thoughts strategy to understand and evaluate visual outputs. Following this, the Q-Bench study [40] presents a binary softmax approach that enables LMMs to compute quantifiable quality scores. This is achieved by utilizing softmax pooling on the logits corresponding to the tokens *good* and *poor*, facilitating a more nuanced assessment of visual quality. Further refining this methodology, Q-Instruct [41] demonstrates the potential of fine-tuning LMMs using a text-based question-answering format, targeting specific low-level visual queries to improve their accuracy in visual quality evaluation. Drawing inspiration from these developments, the Q-Align [42] framework mimics human evaluation mechanisms and post-assessment recalibrations in visual quality scoring, thereby attaining exceptional performance metrics. This progression in research underscores the evolving capabilities of LMMs in the nuanced domain of visual quality assessment, setting a new benchmark for future explorations. These works have fully illustrated the potential of LMMs' quality assessment abilities on 2D media such as images and videos. However, no effort has been put into adapting LMMs to the 3D quality assessment field.

### 2.2 PCQA Development

In the initial phase of PCQA exploration, the MPEG group introduces key Full-Reference PCQA (FR-PCQA) methods such as p2point [27] and p2plane [34], aimed at evaluating point cloud quality. They later present a point-based PSNR-yuv technique to handle colored point clouds [35]. However, these early attempts face some critical challenges in accurately capturing complex distortions through point-level differences. To overcome these challenges mentioned above, more sophisticated FR-PCQA metrics emerge, including PCQM [28], GraphSIM [48] and PointSSIM [2], which integrate structural features to significantly improve performance.

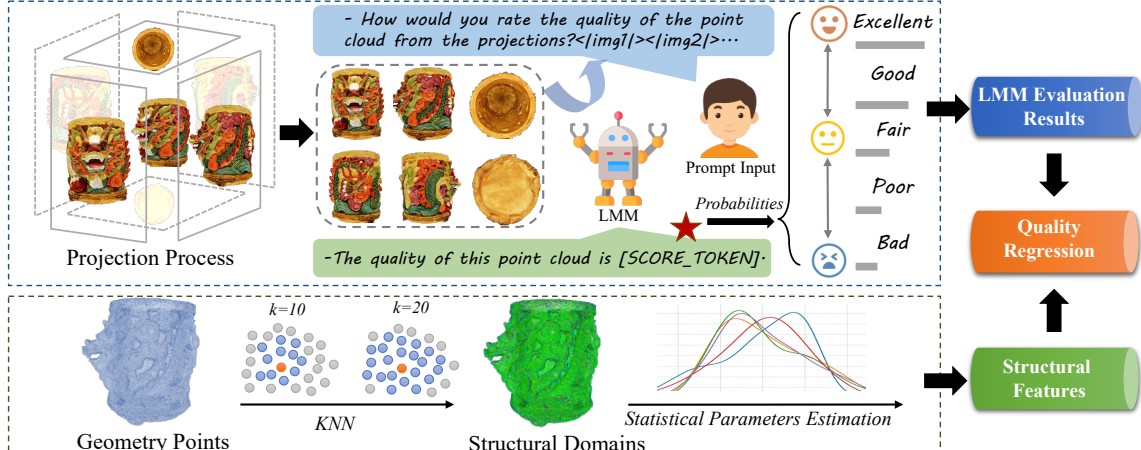

**Figure 2: The framework of the proposed method.**

Inspired by recent advances in no-reference image and video quality assessment (NR-I/VQA) [14, 23, 53, 54, 56, 61], and to cater to a broader range of real-world applications, various NR-PCQA methods have been carried out during the last decade. Chetouani *et al.* [8] apply classical Convolutional Neural Network (CNN) models for quality regression with the assistance of patch-wise handcrafted features. PQA-net [22] adopts multi-view projection for feature extraction, while Zhang *et al.* [58] predict quality-aware parameters by analyzing distributions of geometry and color attributes. Furthermore, Liu *et al.* [25] utilize an end-to-end sparse CNN for direct quality prediction, and Zhou *et al.* [64] focus on structure-guided resampling for extracting relevant features. Some researchers attempt to convert point clouds into videos to employ video quality assessment (VQA) techniques for evaluating perceptual quality [11, 59]. To mitigate computational complexity, GMS-3DQA approach [62] integrates multi-projections into a singular quality map to facilitate feature extraction. Yang *et al.* [47] extend quality assessment methodologies from natural images to point cloud rendering images via domain adaptation techniques. Additionally, Wang *et al.* [38] employs a sophisticated non-local geometry and color gradient aggregation graph model for accurate quality estimation. Furthermore, to provide a more comprehensive learning framework for point clouds from both color and geometry perspectives, MM-PCQA [60] and pmBQA [45] suggest leveraging a multi-modal learning approach to improve PCQA outcomes. In the evolution of PCQA, the potential of text modality for quality representation remains largely untapped. Investigating the approach of leveraging text supervision to facilitate model quality assessment learning emerges as an innovative strategy.

## 3 PROPOSED METHOD

The framework of the proposed method is briefly illustrated in Fig. 2, which includes the LMM evaluation module, the structural feature extraction module, and the quality regression module.

### 3.1 LMM Evaluation

*3.1.1 Projection Acquisition.* Consider a colored point cloud $\mathbf{P} = (p_i, c_i)_{i=1}^{N}$, where $p_i \in \mathbb{R}^{1 \times 3}$ represents the single point consisting of geometry coordinates, $c_i \in \mathbb{R}^{1 \times 3}$ represents the RGB color attributes, and $N$ denotes the total count of points. We then adopt the conventional cube-like viewpoints configuration, widely used in the standard point cloud compression scheme MPEG VPCC [12]. As illustrated in Fig. 2, six orthogonal viewpoints are utilized, each mapping to one of the cube's six faces for generating the projections. For a point cloud $\mathbf{P}$, the rendering process is derived as:

$$\mathbf{I} = \psi(\mathbf{P}),$$
$$\mathbf{I} = \{\mathcal{I}_k | k = 1, \cdots, 6\}, \tag{1}$$

where $\mathbf{I}$ represents the set of the 6 rendered projections and $\psi(\cdot)$ denotes the Open3D-based [63] projections capturing process.

*3.1.2 How to inject PCQA knowledge into LMMs ?* LMM has shown competitive performance in 2D image/video quality assessment [40–42], therefore it is feasible to apply LMM for PCQA tasks by taking the point cloud as a sequence of projections. Then it is vital to solve the core problem of **PCQA Knowledge Injection**. Following the common approaches of training LMMs [18, 49], it is natural to come up with the solution of instructing the LMM with question-answer pairs regarding PCQA issues. Thus we carry out the specific prompt structure as follows:

*-How would you rate the quality of the point cloud from the projections?<|img1|><|img2|>...*
*-The quality of this point cloud is [QA(mos)].*

where *<|img1|><|img2|>...* stands for the image set of projections and *QA(s)* is the **qualitative adjective** of the point cloud which can be obtained from the mean opinion score (MOS) corresponding to the point cloud. Afterward, we can use the designed question-answer pairs to teach LMM PCQA knowledge.

*3.1.3 Transformation from MOS to quality rating.* In daily experiences, humans often give feedback using **qualitative adjectives** *(such as good, bad, superb)* instead of **numerical ratings** *(like 9.2, 2.5, 7.1)*. Hence, implementing visual scoring activities with level-based ratings taps into this *natural tendency* of humans *(to offer qualitative adjectives)*. Similarly, the perception and expression of

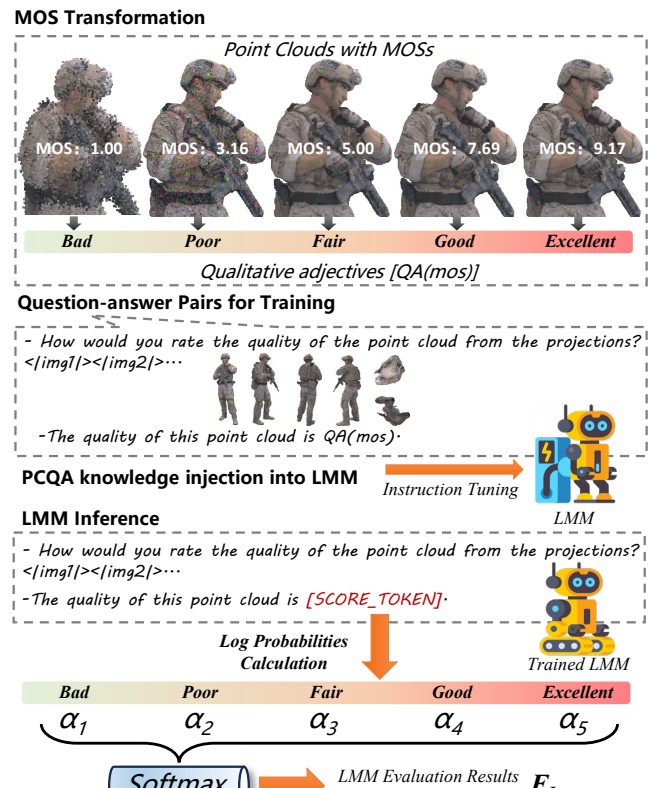

**Figure 3: Illustration of the LLM evaluation pipeline. The point clouds with MOSs are transformed into question-answer pairs for LMM tuning. The LMM evaluation results can be obtained as the set of the probabilities to the predefined qualitative adjectives.**

LMM are akin to humans, which have a better understanding and perception of **qualitative adjectives**. Therefore, converting MOS into corresponding **qualitative adjectives** for its learning is more intuitive than having it learn directly from numbers. Specifically, the transformation of MOS can be achieved by evenly splitting the range from the highest score (M) to the lowest score (m) into five unique intervals, with scores in each interval designated as corresponding quality levels:

$$QA(mos) = w_i \text{ if } m + \frac{i-1}{5} \times (M - m) < mos \leq m + \frac{i}{5} \times (M - m), \quad (2)$$

where $\{w_i|_{i=1}^5\} = \{bad, poor, fair, good, excellent\}$ are the standard text rating levels as defined by ITU [1].

*3.1.4 Obtaining evaluation results via LMM inference.* After training, we can get the evaluation results with the same prompt structure and get the response *[SCORE_TOKEN]* via LMM inference. The *[SCORE_TOKEN]* can be recognized as a log probability map to the **qualitative adjectives**. Then we can compute the final probabilities to the 5-level **qualitative adjectives** from the corresponding log probabilities via softmax as the LMM evaluation results:

$$F_L = \{\frac{e^{\alpha_i}}{\sum_{j=1}^5 e^{\alpha_j}}\}_{i=1}^5, \quad (3)$$

where $\alpha_i$ indicates the log probability of $i$-th **qualitative adjectives** and $F_L$ represents the LMM evaluation results which consist of 5 probabilities after softmax.

## 3.2 Structual Feature Extraction

The inadequacy of single-modality information from projections for point cloud quality evaluation has been established [45, 60]. Therefore, our approach enhances the accuracy of LMM evaluation results (projections only) by integrating geometric structural features, aiming for a more detailed and accurate assessment.

*3.2.1 Structural Domain.* Given the point cloud $\mathbf{P} = (p_i, c_i)_{i=1}^N$, the neighborhood $P_{Nbi}$ of each point $p_i$ can be obtained utilizing the k-nearest neighbors (k-NN) algorithm:

$$\mathbf{P_{Nb}} = \mathbf{KNN}(\mathbf{P}),$$
$$Dist(p, q) = \sqrt{(p_x - q_x)^2 + (p_y - q_y)^2 + (p_z - q_z)^2}, \quad (4)$$

where $N$ represents the total count of points within the point cloud, the term $\mathbf{P_{Nb}}$ denotes the collection of neighborhoods, while $\mathbf{KNN}(\cdot)$ signifies the function of the k-nearest neighbors algorithm. The distance between points $p$ and $q$ is calculated using the Euclidean distance, expressed as $Dist(p, q)$. Given the neighborhood set $P_{Nbi}$ of point $p_i$, we can define the covariance matrix $C_i$ for each point $p_i$, which is characterized by its 3D geometric coordinates:

$$C_i = \frac{1}{K} \sum_{j=1}^K (p_{n_j} - \hat{p})(p_{n_j} - \hat{p})^\top,$$
$$\{p_{n_1}, \cdots, p_{n_K}\} \in P_{Nbi}, \quad (5)$$

where the term $K$ denotes size of neighborhood set $P_{Nbi}$, $p_{n_j}$ is the $j$-th neighboring point in $P_{Nbi}$, $\hat{p}$ is the centroid of this neighborhood, $p_{n_j}$ and $\hat{p}$ are vectors with dimensions $\mathbb{R}^{3 \times 1}$, while $C_i$ is a matrix with dimensions $\mathbb{R}^{3 \times 3}$. Consequently, the eigenvectors for the covariance matrix $C_i$ can be derived as follows:

$$C_i \cdot v_l = \lambda_l \cdot v_l, l \in \{1, 2, 3\}, \quad (6)$$

where $(\lambda_1, \lambda_2, \lambda_3)$ stand for the eigenvalues and $(v1, v2, v3)$ represent the respective eigenvectors, with $\lambda1 > \lambda_2 > \lambda_3$. Consequently, we derive three eigenvalues for each point $p_i$ within the point cloud $\mathbf{P}$. Then we can compute the *linearity* and *planarity* structural domains of the point cloud as:

$$Lin(p_i) = \frac{\lambda_1 - \lambda_2}{\lambda_1},$$
$$Pla(p_i) = \frac{\lambda_2 - \lambda_3}{\lambda_1}, \quad (7)$$

where $Lin(p_i)$ and $Pla(p_i)$ represent the *linearity* and *planarity* values for point $p_i$. The chosen structural domains (*linearity, planarity*) have been demonstrated to exhibit a strong correlation with geometric visual losses, such as compression and downsampling, and have been extensively utilized in numerous PCQA tasks [3, 58, 65].

*3.2.2 Multi-scale Perception.* The multi-scale nature of point cloud visual perception has been noted in the literature [57]. To account for this, we compute structural domains across various scales by varying the scale parameter $k$ in the $\mathbf{KNN}(\cdot)$ process. This approach allows us to derive the multi-scale structural domains as follows:

$$\mathcal{D}_{k=k_{ms}} = \underset{k=k_{ms}}{\mathcal{S}}(\mathbf{KNN}(\mathbf{P})), \mathcal{D} \in \{Lin, Pla\}, \qquad (8)$$

where $\mathcal{D}_{k=k_{ms}}$ denotes the multi-scale structural domains, with $k_{ms}$ represents the set of scale parameters, and $\mathcal{S}(\cdot)$ refers to the process of calculating structural domains as previously described. In our study, we establish the default set of scale parameters as $\{10, 20\}$, signifying that the *linearity* and *planarity* domains are computed using the 10-nearest-neighbor and 20-nearest-neighbor configurations, respectively.

*3.2.3 Statistical Parameters Estimation.* To quantify the quality representation from the structural domains, we employ some basic statistical parameters estimation process:

$$F_S = \{avg(\mathcal{D}_{k=k_{ms}}), std(\mathcal{D}_{k=k_{ms}}), ent(\mathcal{D}_{k=k_{ms}})\}, \\ \mathcal{D} \in \{Lin, Pla\}, \qquad (9)$$

where $avg(\cdot)$, $std(\cdot)$, and $ent(\cdot)$ represent the average function, standard deviation function, and entropy function respectively, and $F_S$ indicates the set of the final extracted structural features.

## 3.3 Quality Regression

To clearly demonstrate the efficacy of the proposed features, we merge the LMM evaluation results with the structural features, and then incorporate them into the visual quality score using support vector regression (SVR):

$$Q = \mathbf{SVR}(F_L \oplus F_S), \qquad (10)$$

where $Q$ indicates the quality values, $\mathbf{SVR}(\cdot)$ represents the SVR regression process, and $\oplus$ denotes the concatenation process.

## 4 EXPERIMENT

### 4.1 Validation Databases

To assess the efficacy of the proposed method, we employ the SJTU-PCQA database [46], the Waterloo point cloud assessment database (WPC)[19], and the WPC2.0 database [20] for validation. The SJTU-PCQA database contains 9 reference point clouds, subjected to seven distortion types (compression, color noise, geometric shift, down-sampling, and three mixed distortions) at six levels, yielding a total of 378 ($9 \times 7 \times 6$) distorted point clouds. The WPC database includes 20 reference point clouds, each modified by four distortions (down-sampling, Gaussian white noise, Geometry-based Point Cloud Compression (G-PCC), and Video-based Point Cloud Compression (V-PCC)), resulting in 740 ($20 \times 37$) distorted point clouds. Meanwhile, the WPC2.0 database features 16 reference point clouds, each undergoing 25 different V-PCC degradation settings, leading to 400 ($16 \times 25$) distorted point clouds.

### 4.2 Implementation Details

*4.2.1 LMM Training.* Following the mainstream choice of LMM-involved quality assessment methods [41–43], we select the mPLUG-Owl-2 [50] as the LMM model in this paper. The model comprises a

CLIP-ViT-Large [32] visual encoder $\mathcal{E}_v$ with 304 million parameters, a visual abstractor $\hat{\mathcal{E}}_v$ with 82 million parameters, and the LLaMA2-7B [36] LLM $\mathcal{L}$ on top of the visual modules, which integrates an additional multi-way module from mPLUG-Owl2, totaling 7.8 billion parameters. Input projections are initially squared through padding before being resized to $448 \times 448$. Let $\mathcal{E}_t$ represent the text embedding layer, with input projections denoted as *<img1><img2>...* and the text prompt as $t$, the detailed formulation of the used LMM model can be expressed as follows:

$$\begin{aligned} \mathcal{H}_v &= \hat{\mathcal{E}}_v(\mathcal{E}_v(< img1 >< img2 > ...)), \\ \mathcal{H}_t &= \mathcal{E}_t(t), \\ \mathcal{H} &= \mathcal{H}_v \oplus \mathcal{H}_t, \\ O &= \mathcal{L}(\mathcal{H}), \end{aligned} \qquad (11)$$

where $\mathcal{H}_v$ and $\mathcal{H}_t$ represent the abstracted tokens for the visual and text input respectively, $O$ stands for the output. For all PCQA databases, the batch size is maintained at 64. The learning rate is fixed at $2 \times 10^{-5}$, with the training process extending over 2 epochs for each variant. We utilize the common GPT [33] loss mechanism, specifically the cross-entropy between the predicted logits and actual labels. The evaluation of performance metrics is conducted using the final weights obtained post-training. Four NVIDIA A100 80G GPUs are employed for the training phase, while a single RTX3090 24G GPU is used to measure inference latency. In the inference stage, only the input texts preceding the *[SCORE_TOKEN]* are inputted into the LMM, leading to the final element of $\mathbf{O}$ representing the targeted probability map.

*4.2.2 Validation Strategy.* Following the methodologies in [4, 11, 60], we utilize a k-fold cross-validation approach in our experiments to ensure a dependable performance evaluation of our proposed method. The SJTU-PCQA, WPC, and WPC2.0 databases consist of 9, 20, and 16 point cloud groups, respectively, leading us to adopt 9-fold, 5-fold, and 4-fold cross-validation for these databases to achieve an approximate 8:2 train-test split. The average of the performance metrics is considered the definitive result. It is crucial to note that the training and testing sets are mutually exclusive to prevent content overlap. For FR-PCQA methods, which do not require training, we evaluate them using the same test sets and report the average performance.

### 4.3 Competitors

17 state-of-the-art quality assessment methods are selected for comparison, which consists of 8 FR-PCQA and 9 NR-PCQA methods:

- The FR-PCQA methods include MSE-p2point (MSE-p2po) [27], Hausdorff-p2point (HD-p2po) [27], MSE-p2plane (MSE-p2pl) [34], Hausdorff-p2plane (HD-p2pl) [34], PSNR-yuv [35], PCQM [28], GraphSIM [48], and PointSSIM [2].
- The NR-PCQA methods include BRISQUE [29], NIQE [30], IL-NIQE [55], IT-PCQA [47], ResSCNN [25], PQA-net [22], 3D-NSS [58], GMS-3DQA [62] and MM-PCQA [60].

Note that BRISQUE, NIQE, IL-NIQE are image-based quality assessment metrics and are validated on the same projections.

**Table 1: Performance on the SJTU-PCQA, WPC, and WPC2.0 databases. Best in red, second in blue.**

| Type | Methods | SJTU-PCQA | | | | WPC | | | | WPC2.0 | | | |
|------|---------|-----------|---|---|---|-----|---|---|---|--------|---|---|---|
| | | SRCC↑ | PLCC↑ | KRCC↑ | RMSE↓ | SRCC↑ | PLCC↑ | KRCC↑ | RMSE↓ | SRCC↑ | PLCC↑ | KRCC↑ | RMSE↓ |
| FR | MSE-p2po | 0.7294 | 0.8123 | 0.5617 | 1.3613 | 0.4558 | 0.4852 | 0.3182 | 19.8943 | 0.4315 | 0.4626 | 0.3082 | 19.1605 |
| | HD-p2po | 0.7157 | 0.7753 | 0.5447 | 1.4475 | 0.2786 | 0.3972 | 0.1943 | 20.8990 | 0.3587 | 0.4561 | 0.2641 | 18.8976 |
| | MSE-p2pl | 0.6277 | 0.5940 | 0.4825 | 2.2815 | 0.3281 | 0.2695 | 0.2249 | 22.8226 | 0.4136 | 0.4104 | 0.2965 | 21.0400 |
| | HD-p2pl | 0.6441 | 0.6874 | 0.4565 | 2.1255 | 0.2827 | 0.2753 | 0.1696 | 21.9893 | 0.4074 | 0.4402 | 0.3174 | 19.5154 |
| | PSNR-yuv | 0.7950 | 0.8170 | 0.6196 | 1.3151 | 0.4493 | 0.5304 | 0.3198 | 19.3119 | 0.3732 | 0.3557 | 0.2277 | 20.1465 |
| | PCQM | 0.8644 | 0.8853 | 0.7086 | 1.0862 | 0.7434 | 0.7499 | 0.5601 | 15.1639 | 0.6825 | 0.6923 | 0.4929 | 15.6314 |
| | GraphSIM | 0.8783 | 0.8449 | 0.6947 | 1.0321 | 0.5831 | 0.6163 | 0.4194 | 17.1939 | 0.7405 | 0.7512 | 0.5533 | 14.9922 |
| | PointSSIM | 0.6867 | 0.7136 | 0.4964 | 1.7001 | 0.4542 | 0.4667 | 0.3278 | 20.2733 | 0.4810 | 0.4705 | 0.2978 | 19.3917 |
| NR | BRISQUE | 0.3975 | 0.4214 | 0.2966 | 2.0937 | 0.2614 | 0.3155 | 0.2088 | 21.1736 | 0.0820 | 0.3353 | 0.0487 | 21.6679 |
| | NIQE | 0.1379 | 0.2420 | 0.1009 | 2.2622 | 0.1136 | 0.2225 | 0.0953 | 23.1415 | 0.1865 | 0.2925 | 0.1335 | 22.5146 |
| | IL-NIQE | 0.0837 | 0.1603 | 0.0594 | 2.3378 | 0.0913 | 0.1422 | 0.0853 | 24.0133 | 0.0911 | 0.1233 | 0.0714 | 23.9987 |
| | IT-PCQA | 0.8651 | 0.8283 | 0.6430 | 1.1661 | 0.4870 | 0.4329 | 0.3006 | 19.8960 | 0.5661 | 0.5432 | 0.3477 | 18.7224 |
| | ResSCNN | 0.8600 | 0.8100 | - | - | - | - | - | - | 0.7500 | 0.7200 | - | - |
| | PQA-net | 0.8372 | 0.8586 | 0.6304 | 1.0719 | 0.7026 | 0.7122 | 0.4939 | 15.0812 | 0.6191 | 0.6426 | 0.4606 | 16.9756 |
| | 3D-NSS | 0.7144 | 0.7382 | 0.5174 | 1.7686 | 0.6479 | 0.6514 | 0.4417 | 16.5716 | 0.5077 | 0.5699 | 0.3638 | 17.7219 |
| | GMS-3DQA | 0.9108 | 0.9177 | 0.7735 | 0.7872 | 0.8308 | 0.8338 | 0.6457 | 12.2292 | 0.8272 | 0.8218 | 0.6277 | 12.9904 |
| | MM-PCQA | 0.9103 | 0.9226 | 0.7838 | 0.7716 | 0.8414 | 0.8556 | 0.6513 | 12.3506 | 0.8023 | 0.8024 | 0.6202 | 13.4289 |
| | **LMM-PCQA(Ours)** | 0.9376 | 0.9404 | 0.8002 | 0.7175 | 0.8825 | 0.8739 | 0.7064 | 11.8171 | 0.8614 | 0.8634 | 0.6723 | 10.6924 |

## 4.4 Evaluation Criteria

Four mainstream evaluation criteria in the quality assessment field are utilized to compare the correlation between the predicted scores and MOSs, which include Spearman Rank Correlation Coefficient (SRCC), Kendall's Rank Correlation Coefficient (KRCC), Pearson Linear Correlation Coefficient (PLCC), Root Mean Squared Error (RMSE). An excellent quality assessment model should obtain values of SRCC, KRCC, PLCC close to 1 and RMSE to 0.

## 4.5 Performance Discussion

The performance comparison between the proposed LMM-PCQA and other PCQA competitors on the SJTU-PCQA, WPC, and WPC2.0 databases is illustrated in Table 1, from which we can draw several conclusions: 1) The LMM-PCQA demonstrates superior performance across all three PCQA databases, even outperforming FR-PCQA methods. For instance, the proposed LMM-PCQA surpasses the second-best NR-PCQA method by about 0.027 (against GMS-3DQA), 0.041 (against MM-PCQA), and 0.034 (against GMS-3DQA) on the SJTU-PCQA, WPC, and WPC2.0 databases from the SRCC values. This highlights LMM's capability to effectively assimilate PCQA knowledge and apply it actively. The consistency in LMM-PCQA's performance across various databases underscores its potential to set new baselines in the PCQA field. 2) All PCQA competitors generally perform better on the SJTU-PCQA database but face significant performance declines on the WPC and WPC2.0 databases. In contrast, LMM-PCQA exhibits the smallest performance drops, with decreases of approximately 0.06 and 0.08 in SRCC values when transitioning from the SJTU-PCQA to the WPC and WPC2.0 databases, respectively. This performance stability underscores LMM-PCQA's robustness and superior ability to handle diverse content effectively, showcasing its adaptability and consistency in quality assessment across different point cloud databases.

**Table 2: Contributions of LMM evaluation results and structural features, where 'w/o LMM' indicates excluding the LMM evaluation results, 'w/o Structural' indicates excluding the structural features.**

| Modal | SJTU-PCQA | | WPC | | WPC2.0 | |
|-------|-----------|---|-----|---|--------|---|
| | SRCC↑ | PLCC↑ | SRCC↑ | PLCC↑ | SRCC↑ | PLCC↑ |
| w/o LMM | 0.6650 | 0.7274 | 0.3598 | 0.3523 | 0.3847 | 0.3951 |
| w/o Structural | 0.9081 | 0.9158 | 0.8488 | 0.8271 | 0.8258 | 0.8381 |
| LMM + Structural | 0.9376 | 0.9404 | 0.8825 | 0.8739 | 0.8614 | 0.8634 |

## 4.6 Ablation Study

*4.6.1 Contributions of LMM evaluation results and structural features.* To fully investigate the contributions and validate the rationality behind the proposed dual streams of features, we decide to undertake an ablation study in this section. The results, as detailed in Table 2, clearly demonstrate that the integration of both feature streams leads to superior performance compared to employing a single feature stream. Upon a detailed examination, it is apparent that LMM evaluation results markedly surpass the structural features in terms of performance. This disparity primarily stems from the fact that some distorted point clouds within the SJTU-PCQA, WPC, and WPC2.0 databases suffer only from color distortions. Structural features, derived from geometric information, inherently lack the capability to recognize these color distortions, resulting in their relatively lower performance in total. However, incorporating structural features enhances the comprehensive understanding of point cloud quality, contributing to the refinement and elevation of the assessment precision in LMM evaluation results. This combined method highlights how important it is to look at features from different angles to fully understand and capture the subtle details of point cloud quality.

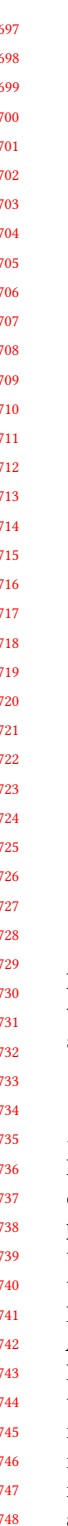

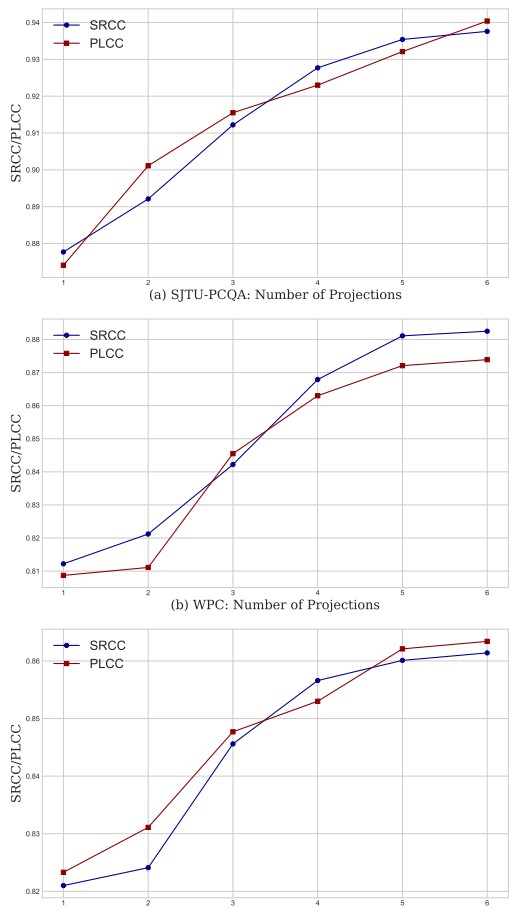

**Figure 4: SRCC/PLCC performance tendency according to the number of used projections on the SJTU-PCQA, WPC, and WPC2.0 databases.**

*4.6.2 Influence of the number of projections.* The proposed LMM-PCQA utilizes 6 projections as default. In this section, we further change the number of used projections to test the corresponding performance influence. Specifically, we randomly select 1-6 projections from the cube-like 6 projections setting as the input projections of LMM-PCQA. The performance tendency is illustrated in Fig. 4, from which we can draw several interesting conclusions: 1) As the number of projections increases, the performance of LMM-PCQA also improves correspondingly, indicating that increasing the number of projections can encompass more effective quality information, thereby enhancing the final performance. 2) Specifically, when the number of projections increases from 2 to 5, the improvement is significantly more pronounced. This suggests that at this stage, the quality information is not yet redundant, and the benefit of increasing the number of projections is relatively large. However, when the number of projections increases from 5 to 6, the performance improvement is relatively low, indicating that the quality information has become somewhat saturated, and further increases in the number of projections yield diminishing returns.

**Table 3: Performance of the multi-scale structural features, where k is the scale parameter of the KNN algorithm.**

| Model | SJTU-PCQA | | WPC | | WPC2.0 | |
|---|---|---|---|---|---|---|
| | SRCC↑ | PLCC↑ | SRCC↑ | PLCC↑ | SRCC↑ | PLCC↑ |
| w/o LMM | | | | | | |
| k=10 | **0.6090** | **0.6584** | **0.3261** | **0.3482** | **0.3366** | 0.2777 |
| k=20 | 0.5920 | 0.6311 | 0.1795 | 0.2720 | 0.3224 | **0.3129** |
| k=10,20 | **0.6650** | **0.7274** | **0.3598** | **0.3523** | **0.3847** | **0.3951** |
| with LMM | | | | | | |
| k=10 | 0.9140 | 0.9176 | **0.8564** | **0.8554** | 0.8432 | 0.8466 |
| k=20 | **0.9199** | **0.9179** | 0.8466 | 0.8488 | **0.8578** | **0.8562** |
| k=10,20 | **0.9376** | **0.9404** | **0.8825** | **0.8739** | **0.8614** | **0.8634** |

**Table 4: The cross-database evaluation performance, 'WPC→SJTU-PCQA' signifies that the model is trained using the WPC database and tested according to the standard testing protocol of the SJTU-PCQA database. We eliminate those point cloud groups from the WPC database that have reference counterparts in the WPC2.0 testing sets, thereby preventing content duplication.**

| Model | WPC→SJTU-PCQA | | WPC→WPC2.0 | |
|---|---|---|---|---|
| | SRCC↑ | PLCC↑ | SRCC↑ | PLCC↑ |
| PQA-net | 0.5411 | 0.6102 | 0.6006 | 0.6377 |
| 3D-NSS | 0.1817 | 0.2344 | 0.4933 | 0.5613 |
| GMS-3DQA | 0.7421 | 0.7611 | 0.7822 | 0.7714 |
| MM-PCQA | **0.7991** | **0.7902** | **0.7917** | **0.7935** |
| LMM-PCQA(Ours) | **0.8246** | **0.7999** | **0.8385** | **0.8387** |

*4.6.3 Effect of the multi-scale structural features.* To quantify the contributions of the multi-scale mechanism, we validate the performance of structural features with different scale parameters under two settings: with LMM evaluation results and without LMM evaluation results. The experimental performance is listed in Table 3. From the table, we can find that the multi-scale structural features with k=10,20 perform better than the single-scale features whether the LMM evaluation results are involved or not, which confirms the effectiveness of the proposed multi-scale mechanism. This can be attributed to that humans tend to perceive the visual quality of point clouds from a multi-scale perspective.

## 4.7 Cross-database Validation

To the generalization ability of the proposed LMM-PCQA, we conduct the cross-database validation in this section. Considering that the SJTU-PCQA, WPC, and WPC2.0 databases contain 378, 740, and 400 distorted point clouds respectively, we pre-train LMM-PCQA on the WPC database (largest in scale) and validate the performance on the SJTU-PCQA and WPC2.0 databases (smaller in scale). The competitive NR-PCQA methods (PQA-net, 3d-NSS, GMS-3DQA, and MM-PCQA) are included for comparison. The experimental performance is shown in Table 4, from which we can make several observations: 1) The proposed LMM-PCQA achieves the best cross-database validation performance against all competitors, which confirms the strong generalization ability of LMM-PCQA. 2) Most methods obtain higher WPC→WPC2.0 performance

**Table 5: Distortion-specific performance results on the SJTU-PCQA database, where OT represents octree-based compression, CN represents color noise, DS represents down-sampling, DS+CN represents down-sampling and color noise, DS+GGN represents down-sampling and geometry Gaussian noise, GGN represents geometry Gaussian noise, and CN+GGN represents color noise and geometry Gaussian noise respectively.**

| Distortion | OT | | CN | | DS | | DS+CN | | DS+GGN | | GGN | | CN+GGN | |
|---|---|---|---|---|---|---|---|---|---|---|---|---|---|---|
| Method | SR↑ | PL↑ | SR↑ | PL↑ | SR↑ | PL↑ | SR↑ | PL↑ | SR↑ | PL↑ | SR↑ | PL↑ | SR↑ | PL↑ |
| MSE-p2po | 0.71 | 0.76 | nan | nan | 0.93 | 0.95 | **0.96** | 0.87 | 0.96 | 0.89 | **0.98** | 0.89 | **0.99** | 0.90 |
| HD-p2po | 0.64 | 0.69 | nan | nan | 0.82 | 0.88 | 0.80 | 0.75 | 0.94 | 0.91 | **0.98** | 0.91 | **0.99** | 0.91 |
| MSE-p2pl | 0.55 | 0.62 | nan | nan | 0.87 | 0.92 | 0.85 | 0.81 | 0.96 | 0.75 | **0.97** | 0.85 | **0.98** | 0.86 |
| HD-p2pl | 0.54 | 0.58 | nan | nan | 0.82 | 0.87 | 0.81 | 0.79 | 0.94 | 0.77 | 0.95 | 0.88 | 0.97 | 0.85 |
| PSNR-yuv | 0.59 | 0.54 | 0.86 | 0.87 | 0.91 | 0.91 | **0.96** | 0.91 | **0.97** | 0.94 | **0.98** | 0.95 | **0.99** | 0.96 |
| PCQM | 0.80 | 0.84 | 0.86 | 0.85 | 0.93 | 0.96 | **0.97** | **0.94** | 0.96 | 0.90 | **0.98** | 0.93 | **0.99** | 0.93 |
| GraphSIM | 0.82 | 0.81 | 0.82 | 0.90 | **0.96** | **0.97** | 0.91 | **0.95** | 0.95 | 0.95 | 0.96 | **0.97** | 0.97 | **0.98** |
| PointSSIM | 0.80 | **0.88** | 0.87 | 0.87 | 0.93 | 0.93 | **0.97** | 0.93 | 0.96 | **0.97** | **0.98** | **0.97** | **0.99** | 0.96 |
| PQA-net | 0.81 | 0.82 | 0.84 | 0.83 | 0.91 | 0.92 | 0.93 | 0.91 | 0.89 | 0.89 | 0.95 | **0.96** | 0.97 | 0.96 |
| 3D-NSS | 0.60 | 0.67 | 0.85 | 0.79 | 0.80 | 0.84 | 0.94 | 0.93 | 0.90 | 0.90 | 0.96 | 0.93 | **0.98** | 0.94 |
| GMS-3DQA | 0.83 | 0.84 | 0.91 | 0.92 | 0.95 | 0.95 | 0.95 | 0.93 | 0.96 | **0.97** | **0.97** | 0.93 | **0.98** | 0.95 |
| MM-PCQA | **0.84** | 0.83 | **0.92** | **0.93** | 0.94 | 0.96 | 0.94 | **0.95** | 0.94 | 0.93 | 0.95 | 0.94 | **0.98** | 0.96 |
| LMM-PCQA | **0.89** | **0.90** | **0.97** | **0.97** | **0.97** | **0.98** | 0.95 | **0.94** | **0.98** | **0.98** | 0.96 | 0.93 | **0.98** | **0.97** |

**Table 6: Distortion-specific performance results on the WPC database, where DS represents down-sampling, GN represents geometry and color Gaussian noise, G-PCC represents geometry-based compression, and V-PCC represents video-based compression.**

| Distortion | DS | | GN | | G-PCC | | V-PCC | |
|---|---|---|---|---|---|---|---|---|
| Method | SR↑ | PL↑ | SR↑ | PL↑ | SR↑ | PL↑ | SR↑ | PL↑ |
| MSE-p2po | 0.46 | 0.46 | 0.67 | 0.63 | 0.84 | 0.72 | 0.41 | 0.44 |
| HD-p2po | 0.39 | 0.35 | 0.69 | 0.70 | 0.80 | 0.70 | 0.36 | 0.36 |
| MSE-p2pl | 0.41 | 0.40 | 0.61 | 0.47 | 0.61 | 0.48 | 0.42 | 0.40 |
| HD-p2pl | 0.40 | 0.41 | 0.62 | 0.51 | 0.57 | 0.59 | 0.41 | 0.41 |
| PSNR-yuv | 0.23 | 0.28 | 0.79 | 0.88 | 0.47 | 0.43 | 0.44 | 0.48 |
| PCQM | 0.66 | 0.64 | **0.89** | **0.89** | 0.86 | 0.77 | 0.83 | 0.78 |
| GraphSIM | 0.56 | 0.57 | 0.79 | 0.81 | 0.75 | 0.74 | 0.71 | 0.70 |
| PointSSIM | 0.35 | 0.34 | 0.83 | 0.87 | **0.91** | **0.95** | 0.51 | 0.41 |
| PQA-net | 0.61 | 0.63 | 0.77 | 0.78 | 0.87 | 0.88 | 0.76 | 0.77 |
| 3D-NSS | 0.55 | 0.51 | 0.81 | 0.83 | 0.86 | 0.87 | 0.49 | 0.47 |
| GMS-3DQA | 0.72 | 0.73 | 0.87 | 0.88 | 0.89 | 0.89 | 0.86 | **0.88** |
| MM-PCQA | **0.75** | **0.74** | 0.88 | 0.87 | 0.88 | 0.89 | **0.89** | 0.86 |
| LMM-PCQA | **0.76** | **0.79** | **0.91** | **0.91** | **0.96** | **0.96** | **0.90** | **0.89** |

than WPC→SJTU-PCQA performance. This might be because the WPC2.0 database contains only compression distortions, which undergo part of a similar distortion generation process of the WPC database. Therefore, the quality representation learned from the WPC database is more effective on the WPC2.0 database.

### 4.8 Distortion-specific Evaluation

To verify the effectiveness of the proposed LMM-PCQA on different kinds of distortions, we carry out the distortion-specific evaluation experiment in this section. The performance comparison is shown in Table 5 (The 'nan' values for MSE-p2po, HD-p2po, MSE-p2pl, and HD-p2pl arise because these methods only analyze the geometric differences between the reference and distorted point clouds,

thereby failing to account for color distortions.) and Table 6, from which we can obtain several findings: 1) The proposed LMM-PCQA achieves the best performance on 4 of 7 distortion types and all distortion types on the SJTU-PCQA database and the WPC database respectively, which suggests that LMM-PCQA is effective at dealing with various kinds of distortions. 2) Although LMM-PCQA does not achieve the top performance on DS+CN, GGN, and CN+GGN distortions within the SJTU-PCQA database, the performance gap to the best is minimal, with a difference of no more than 0.02 in terms of SRCC values. This suggests that LMM-PCQA remains in the top tier. 3) Upon examining Table 6 more closely, it is evident that all PCQA methods exhibit substantial declines in performance with the 'DS' distortion. The underlying reason for this trend is the simplicity of the point cloud reference models used in the WPC database, leading to a reduced sensitivity to downsampling distortion.

## 5 CONCLUSION

In conclusion, this paper pioneers the integration of LMMs with PCQA, unveiling the untapped potential of LMMs in this domain. Our research successfully demonstrates the feasibility of adapting LMMs for PCQA through text supervision, enhancing their capability to evaluate 3D visual quality from 2D projections. We also propose to capture multi-scale structural features to offer a more holistic view of the point cloud quality. By combining LMM evaluation results and structural features, our approach significantly improves the accuracy of PCQA. We carry out thorough experiments to demonstrate the effectiveness of the proposed LMM-PCQA, as well as its robustness and ability to generalize across various distortion types and diverse point cloud content. The encouraging results not only validate our methodology but also lay the groundwork for future research. We hope our work will serve as a stepping stone for further exploration into the synergy between LMMs and PCQA, driving forward innovations in 3D visual quality analysis.

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
