# OpenReview forum: "LMM-PCQA: Assisting Point Cloud Quality Assessment with LMM"
_acmmm.org/ACMMM/2024/Conference — MM2024 Oral_

### Official Review · Reviewer_rrPg · 2024-05-20

**Rating:** 5
**Confidence:** 3

**Summary:**

The paper introduces *LMM-PCQA*, which utilizes both large multi-modal models (LLMs) and multi-scale geometry feature extraction to tackle the challenge of point cloud quality assessment (PCQA). The experimental results show the effectiveness of LMMs on this task and validate the rationality of employing the proposed modules. In general, the method is quite interesting and experiment is solid.

**Strengths:**

>1. The idea of employing LMM in PCQA task is the first trial in this field. The framework is well designed, which helps expand the quality assessment applications of LMMs. Aligning quality levels of the point clouds with text descriptions is interesting and novel.
>2. The multi-scale geometry feature extraction is proposed to make up the disadvantages of LMMs, which is quite suitable in this framework.

The paper is well-written and can provide inspiration for further studies.

**Limitations:**

>1. Since training LMMs is quite expensive and consumes large GPU resources, I want to know whether the authors will release the LMM weights. It is very important to the community.
>2. More details are needed in the Sec. 3.1.1. How do the authors define the camera parameters such field of view and viewing distances. Moreover, in Figure 4,  please further explain the curve tendency regarding number of the projections.
>3. It is very interesting to classify the quality levels rather than directly regression with LMMs. Is there any insight or theory behind this trial? Please explain.

**Suitability:**

3

---

### Official Review · Reviewer_wGkZ · 2024-05-20

**Rating:** 4
**Confidence:** 3

**Summary:**

This work investigates the feasibility of imparting Point Cloud Quality Assessment (PCQA) knowledge to LMMs through text supervision. They implement this by firstly fine-tuning LLM from 2D rating logins projected from point clouds. After that, by combining LMM evaluation results and proposed multi-scale structural features, LMM-PCQA demonstrates exceptional performance across various PCQA databases.

**Strengths:**

* The idea of this paper is interesting, exploring the quality representation potential of LMM on 3D point clouds.
* Experiments in this paper are well designed.

**Limitations:**

* The writing of the paper could be sometimes misleading. Some key information, such as mean opinion score (MOS) in S3.1, a qualitative measurement of point cloud is missing, how is it calculated? Furthermore, the format of text makes the essay hard to read, such as bold text, italics. For example, from L122-139, the paragraph is kinda sloppy, and with specific font in Line 135, not sure if it is deliberate to do so and why.

* The performance increase is likely small. One example is in discussion from L628 to L637, the stability analysis of algorithm across various dataset is likely less convincing. As we can see from Table 1, LMM-PCQA exhibits performance drops with decreases of approximately 0.06 and 0.08 in SRCC values when transitioning from the SJTU-PCQA to the WPC and WPC2.0 databases, while GMS-3DQA decreases 0.08 and 0.09, MM-PCQA 0.07 and 0.09 respectively. Furthermore, it is interesting to analyse why FR-Type methods (like PointSSIM, GraphSIM) shows worse robustness to handle diverse content.

**Suitability:**

3

---

### Official Review · Reviewer_zR7p · 2024-05-21

**Rating:** 3
**Confidence:** 3

**Summary:**

In this paper, the authors claimed that they investigate the feasibility of imparting Point Cloud Quality Assessment (PCQA) knowledge to large multi-modality models (LMMs) through text supervision and proposed a PCQA pipeline named LMM-PCQA. In addition, they extracted multi-scale structural features of point cloud to helps LMM gain a more comprehensive understanding of point cloud visual quality. And experimental results affirm the effectiveness of LMM-PCQA.

**Strengths:**

The article provides a clear introduction to the methodology, and the corresponding experimental results substantiate the contributions mentioned in the paper.

**Limitations:**

1. The main contribution of this paper lies in the introduction of LMMs for performing PCQA. The idea is straightforward, and the proposed pipeline in the paper simply combines structural features with the inference results of the LMMs in a loosely coupled manner, which lacks novelty.

2. The paper using the structural features of point cloud to assist the decision-making of LMMs, and the corresponding ablation experiments demonstrate that utilizing only LMMs does not achieve the best performance. These raises the question of whether the limitation lies in the inherent constraints of the large language model itself or in the methodology employed.

3. It is advised that the author considers tightly integrating the structural features within the inference process of the LMMs, rather than simply replacing traditional modules with a LMMs.

**Suitability:**

3

---

### Official Review · Reviewer_Up5a · 2024-05-27

**Rating:** 5
**Confidence:** 4

**Summary:**

In this paper, a point cloud quality assessment metric is proposed that relies on quality ratings from a large multi-modality model (LMM) on multiple 2D projections of the 3D point cloud, and multi-scale structural features computed on the 3D point cloud itself. The authors claim that this is the first attempt to use LMMs for the quality evaluation of 3D models and, specifically, for point clouds.

**Strengths:**

+ timely topic
+ well-written
+ sufficient analysis

The paper is generally good. The topic is nicely motivated and the paper is well-structured and mostly clear (some details are missing; see below). The performance evaluations are sufficient, although some extra results could be added (see below).

**Limitations:**

- some clarifications and further details are needed

In section 3.1.1, it is written:
"We then adopt the conventional cube-like viewpoints configuration, widely used in the standard point cloud compression scheme MPEG VPCC [12]."
In VPCC, there is no cube-like projection process, as the one indicated in Figure 2. Instead, the projection process is patch-based and a bit more complex. The provided visual example resembles more approaches such as the ones discussed in [1,2] (from the references below) and [46] (from the paper's references). Please rephrase as needed.

In section 3.1.1, please provide some further information regarding the projection process. Specifically, do the authors consider the distance of the virtual camera from the model or apply any model scaling (i.e., could be potentially used to fit the projected model into certain 2D boundaries) in the "Open 3D-based projections" of the point clouds? What if the projected region of the model is too small compared to the background region, or too large that shows only a fraction of the model? How is the LMM model affected by such variations/configurations? On the same note, I think that it would make the related work more complete if the authors would refer to some projection-based point cloud quality assessment methods in section 2.2, such as [1,2] (from the references below) and [46] (from the paper's references).

In section 3.2.3, I understand that the average, standard deviation, and entropy functions are applied to structural feature values obtained for all points of a point cloud. Is this correct? Maybe the authors could make explicit in the text that all points are used for the computation of the structural domain (in 3.2.1) and the statistical parameters estimation (in 3.2.3).

In section 4.2.2, it is written:
".. that the training and testing sets are mutually exclusive to prevent content overlap."
Please clarify in the text whether distorted versions of the same content are used for both training and testing. I assume not, but this is not explicit.

In section 4.6.2, it is written:
"Specifically, we randomly select 1-6 projections from the cube-like 6 projections setting as the input projections of LMM-PCQA."
I understand that in the results shown in Figure 4 for 1 projection, the authors randomly chose 1 out of the 6 projections per stimulus, for both training and testing sets. Is this correct? If yes, what is the reason for this choice? The authors could also comment on the effect of this choice on the obtained results.

In section 4.7, the same experiment could be performed using SJTU and WPC2.0 for training (and testing on the remaining two).

In section 4.8, the authors chose SJTU and WPC and not WPC2.0 in this experiment. Is there a reason for this? Please add a sentence explaining that, or add corresponding results for WPC2.0.

[1] https://doi.org/10.1117/12.2322741
[2] https://doi.org/10.1109/QoMEX.2019.8743277

**Suitability:**

3

---

### Meta-Review · Area_Chair_f1Q1 · 2024-07-02

**Recommendation:** Accept (Oral)
**Confidence:** 4

**Metareview:**

This paper was reviewed by four experts in the field. The recommendations were mixed initially, but after authors’ feedback and discussion, the reviewers reached a consensus of acceptance, including Accept, Weak Accept, Borderline Accept, Borderline Accept. ACs also agreed this is an interesting work with solid experimental support. Based on this, the decision is to recommend the paper for acceptance to ACM Multimedia 2024.

Still, we recommended the authors to carefully read all reviewers’ final feedback, and revise the manuscript as needed. We congratulate the authors on the acceptance of their paper!